# Deep learning enables accurate clustering with batch effect removal in single-cell RNA-seq analysis

Xiangjie Li[1,2,3], Kui Wang[1,4], Yafei Lyu[1], Huize Pan[5], Jingxiao Zhang[2], Dwight Stambolian[6], Katalin Susztak[7], Muredach P. Reilly[5], Gang Hu [1,8 ✉] & Mingyao Li [1 ✉]

Single-cell RNA sequencing (scRNA-seq) can characterize cell types and states through unsupervised clustering, but the ever increasing number of cells and batch effect impose computational challenges. We present DESC, an unsupervised deep embedding algorithm that clusters scRNA-seq data by iteratively optimizing a clustering objective function. Through iterative self-learning, DESC gradually removes batch effects, as long as technical differences across batches are smaller than true biological variations. As a soft clustering algorithm, cluster assignment probabilities from DESC are biologically interpretable and can reveal both discrete and pseudotemporal structure of cells. Comprehensive evaluations show that DESC offers a proper balance of clustering accuracy and stability, has a small footprint on memory, does not explicitly require batch information for batch effect removal, and can utilize GPU when available. As the scale of single-cell studies continues to grow, we believe DESC will offer a valuable tool for biomedical researchers to disentangle complex cellular heterogeneity.

[1] Department of Biostatistics, Epidemiology and Informatics, Perelman School of Medicine, University of Pennsylvania, Philadelphia, PA 19104, USA. [2] Center for Applied Statistics, School of Statistics, Renmin University of China, Beijing 100872, China. [3] State Key Laboratory of Cardiovascular Disease, Fuwai Hospital, National Center for Cardiovascular Diseases, Chinese Academy of Medical Sciences and Peking Union Medical College, Beijing 100037, China. [4] Department of Information Theory and Data Science, School of Mathematical Sciences and LPMC, Nankai University, Tianjin 300071, China. [5] Division of Cardiology, Department of Medicine, Columbia University Medical Center, New York, NY 10032, USA. [6] Department of Ophthalmology, Perelman School of Medicine, University of Pennsylvania, Philadelphia, PA 19104, USA. [7] Departments of Medicine and Genetics, Perelman School of Medicine, University of Pennsylvania, Philadelphia, PA 19104, USA. [8] School of Statistics and Data Science, Key Laboratory for medical Data Analysis and Statistical Research of Tianjin, Nankai University, Tianjin 300071, China. ✉email: huggs@nankai.edu.cn; mingyao@pennmedicine.upenn.edu

A primary challenge in scRNA-seq analysis is analyzing the ever increasing number of cells, which can be thousands to millions in large projects such as the Human Cell Atlas[1]. Identifying cell populations is challenging in large datasets because many existing scRNA-seq clustering methods cannot be scaled up to handle them. Large scRNA-seq datasets often include cells that are easy to cluster, and it is desirable to learn expression patterns from these cells because they provide valuable information on cluster-specific gene expression signatures. These cells can further help improve clustering of cells that are hard to cluster. As the number of cells grows in scRNA-seq studies, another major challenge in analysis is batch effect, which is systematic gene expression difference from one batch to another[2]. Batch effect is inevitable in studies involving human tissues because the data are often generated at different times and the batches can confound biological variations. Failure to remove batch effect will complicate downstream analysis and lead to a false interpretation of results.

ScRNA-seq clustering and batch effect removal are typically addressed through separate analyses. Commonly used approaches to remove batch effect include canonical correlation analysis (CCA)[3], mutual nearest neighbors (MNN) approach[4], and the combination of MNN and CCA as implemented in Seurat 3.0[5]. After batch effect removal, clustering analysis is performed to identify cell clusters using methods such as Louvain's method[6], Infomap[7], shared nearest neighbors[8], or consensus clustering with SC3[9]. However, some studies might deplete or enrich certain cell types, which can lead to cell-type-specific batch effect[10]. Even when processed together, some cell types might be more vulnerable to batch effect than others. Haghverdi et al.[4] found that consideration of cell-type-specific batch effects rather than a globally constant batch effect for all cells leads to improved batch effect removal.

Methods such as CCA, MNN, and Seurat 3.0 are based on pairwise analysis in which cells from two batches are considered at a time. For data with more than two batches, the first batch in order will be used as the reference batch to correct cells in the second batch, and the corrected values of the second batch are then added to the reference batch. This procedure is repeated until cells in all batches are corrected. As a pairwise procedure, the order in which batches are corrected will affect the final results. Scanorama[11] is also pairwise based, although it does not require a reference batch. Other methods such as scVI[12] and BERMUDA[13] explicitly incorporate batch information in analysis, thus can jointly analyze cells from all batches simultaneously.

Since clustering and batch effect removal are interrelated, an ideal approach for batch effect removal should be performed jointly with clustering. It is also desirable to have a method that can simultaneously include cells from all batches in the analysis. Here we present DESC, an unsupervised deep learning algorithm that iteratively learns cluster-specific gene expression representation and cluster assignments for scRNA-seq analysis. DESC gradually removes batch effect over iterations, as long as technical differences across batches are smaller than true biological variations (e.g., between cell types). We compared DESC with many state-of-the-art methods for scRNA-seq analysis, including CCA, MNN, Seurat 3.0, scVI, BERMUDA, and scanorama. Through comprehensive analyses of datasets with various degrees of complexities, we show that DESC is able to remove complex batch effect, preserve biological variations, and can reveal both discrete and pseudotemporal structure of cells.

## Results

### Methods overview.
An overview of DESC is shown in Fig. 1a. Using a deep neural network, DESC initializes parameters obtained from an autoencoder and learns a nonlinear mapping function from the original scRNA-seq data space to a low-dimensional feature space by iteratively optimizing a clustering objective function. This iterative procedure moves each cell to its nearest cluster centroid, balances biological and technical differences between clusters, and gradually reduces the influence of batch effect. DESC also enables soft clustering by assigning cluster-specific probabilities to each cell, which facilitates the clustering of cells with high confidence. The resulting maximum probabilities for cluster assignment also provide valuable information on cell transition for cells that originate from a genuinely continuous process.

### Application to macaque retina data with complex batch effect.
To evaluate the performance of DESC, we analyzed a scRNA-seq dataset that includes 21,017 foveal and 9285 peripheral bipolar cells from retina in four macaques[10]. Since 80% of total cells in peripheral retina are rod photoreceptors, rods were depleted (anti-CD73) and retinal ganglion cells were enriched (CD90+) from peripheral samples before profiling. This dataset is complex in that there are three levels of batch effect: animal level (four macaques), region level (fovea and periphery), and sample level (30 samples across macaques and regions). Peng et al.[10] found that CCA and MNN cannot completely remove batch effect, collapsed distinct cell types, or have varying impact on different cell classes. For this complex dataset, DESC is effective in removing batch effect and yields high clustering accuracy (adjusted Rand index (ARI) 0.919–0.970) (Fig. 1b–e). The cells are mixed well regardless whether sample, region, or animal was used to define batch in gene expression standardization, indicating the robustness of DESC. In contrast, CCA, MNN, Seurat 3.0, scVI, BERMUDA, and scanorama are all sensitive to batch definition, and the cells are separated by sample when region or animal was used to define batch in analyses (Fig. 2 and Supplementary Note 2).

In many studies, particularly those involving human tissues, batch effect might be confounded with biological conditions. To reduce the risk of removing true biological variations, it is desirable to have a method that is agnostic to batch. To this end, we further analyzed the data without using any batch information. Among all methods we evaluated, only scVI allows the analysis without the specification of batch in analysis. Figure 3a shows that DESC yields high clustering accuracy (ARI 0.920) with the cells well mixed when batch information was not utilized in the analysis. However, cells from different samples are completely separated in scVI when batch information was not provided (Fig. 3b), and the ARI dropped to 0.242 (Fig. 1e), indicating its strong reliance on how batch is defined in the analysis.

### Application to pancreatic islet data from four protocols.
Next, we tested whether DESC can remove batch effect for data generated through different scRNA-seq protocols. We combined four publicly available datasets on human pancreas generated using Fluidigm C1[14], SMART-seq2[15], CEL-seq[16], and CEL-seq2[17]. Figure 4a shows that cells in these four datasets mixed well in DESC representation, and DESC yields the highest ARI value (0.945), which is much higher than MNN (0.629), scVI (0.696), BERMUDA (0.484), and scanorama (0.537) (Fig. 4b). Although Seurat 3.0 also has high ARI (0.896), its classification accuracies for α and β cells are only 92.1% and 88.1%, respectively, whereas the accuracies are 96.5% for α cells and 98.3% for β cells in DESC, respectively, (Supplementary Note 3). We also note that 2.32% of the α cells and 11.3% of the β cells were misclassified as a single cluster in Seurat 3.0, even though α and β cells are known to have distinct gene expression profiles (Supplementary Fig. 5). Seurat 3.0 relies on anchor cells between pairs of datasets, which are

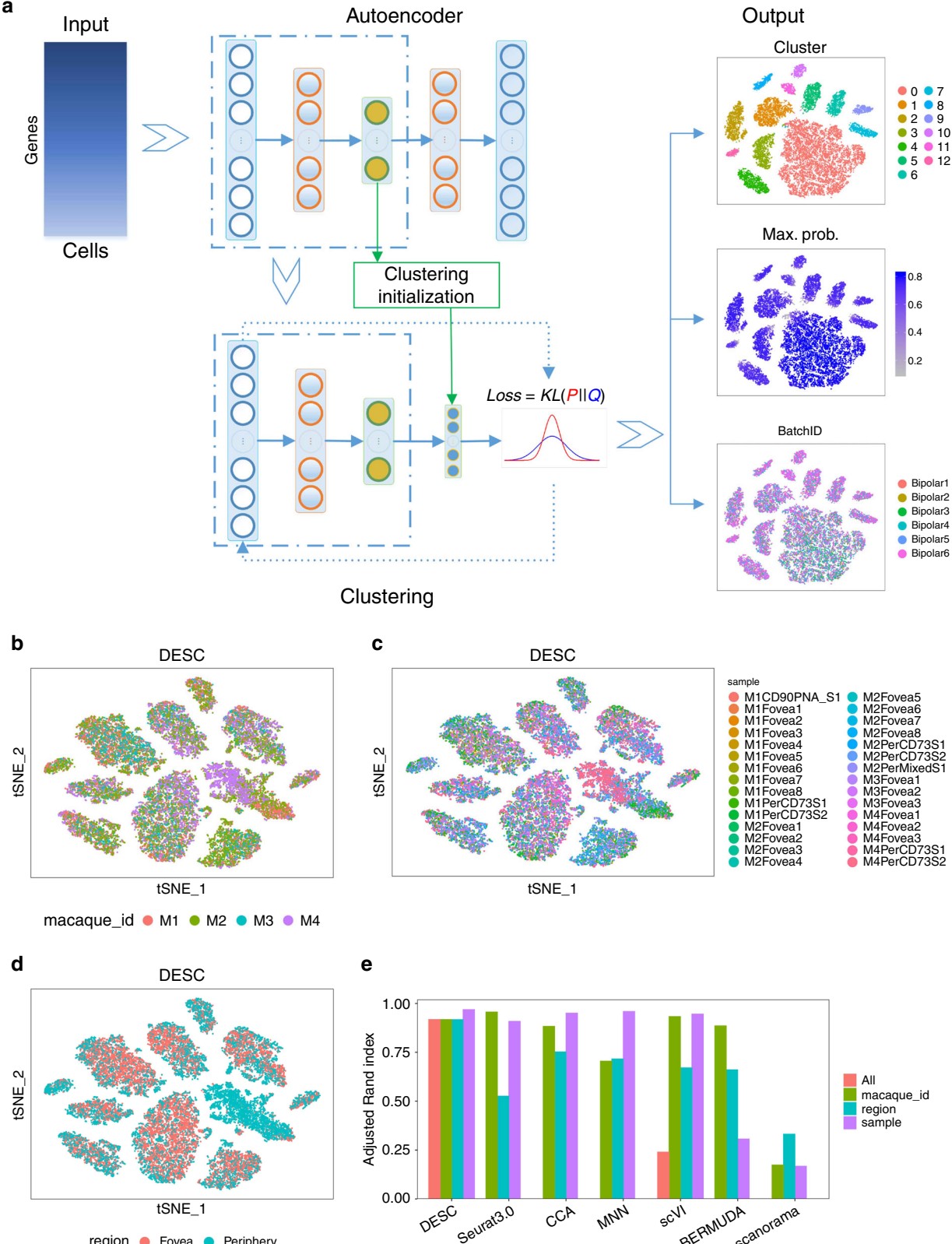

**Fig. 1 The workflow of DESC. a** Overview of the DESC framework. DESC starts with parameter initialization in which a stacked autoencoder is used for pretraining and learning a low-dimensional representation of the input gene expression matrix. The resulting encoder is then added to the iterative clustering neural network to cluster cells iteratively. The final output of DESC includes cluster assignment, the corresponding probabilities for cluster assignment for each cell, and the low-dimensional representation of the data; **b–d** The t-SNE plots of DESC for the macaque retina scRNA-seq data generated by Peng et al.[10] The plots are colored by macaque id (**b**), sample id (**c**), and region (**d**). **e** The ARIs of different methods. The ARIs were calculated when taking different information (macaque id, sample id, region id) as batch in analysis, and "All" was calculated when no batch information was provided in analysis.

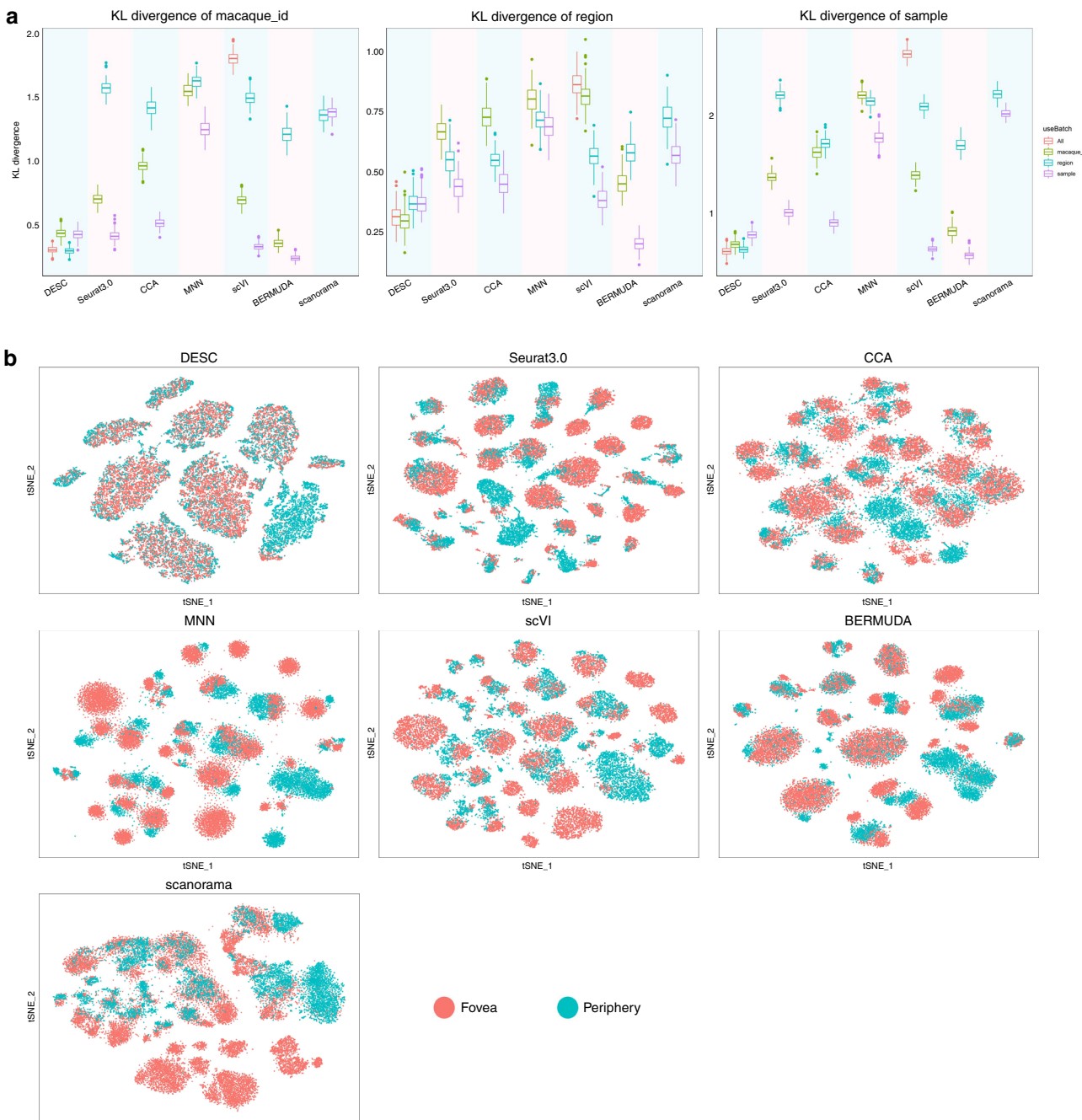

**Fig. 2 Comparison of the robustness of different methods for batch definition based on the macaque retina scRNA-seq data. a** The KL divergences calculated for macaque id (left plot), region id (middle plot), and sample id (right plot) when taking macaque id, region id, or sample id as the batch definition in analysis for each method. The box represents the interquartile range, the horizontal line in the box is the median, and the whiskers represent the 1.5 times interquartile range. The word "All" in legend for DESC and scVI indicates that they take the whole dataset as input without considering any batch information in analysis. This figure shows that DESC yields robust results for batch effect removal no matter what batch information was provided in analysis. However, other methods are sensitive to the choice of batch definition. **b** The t-SNE plots showing region distribution for different methods when region was treated as batch in analysis.

hypothesized to originate from the same biological state, to remove batch effect. However, misidentification of anchors from different batches might have led to reduced accuracy for the α and β cells' classification.

The main idea of DESC is to use "easy-to-cluster" cells to guide the neural network to learn cluster-specific gene expression features, while ignoring other unwanted noises such as batch effect. Specifically, the auxiliary distribution **P** gives cells near the cluster centroid, i.e., easy-to-cluster cells, higher probabilities, in

the process of optimizing the Kullback–Leibler (KL) divergence between distributions **P** and **Q**. By doing this, DESC ignores information unrelated to cell clustering by adjusting the network weight using the gradient descent algorithm, and learns cluster-specific information. Since DESC learns information on cell clusters from those "easy-to-cluster cells," while ignoring other irrelevant information by constructing the auxiliary distribution **P** and optimizing the KL divergence between **P** and **Q**, thus as long as technical differences (e.g., between batches) are smaller than

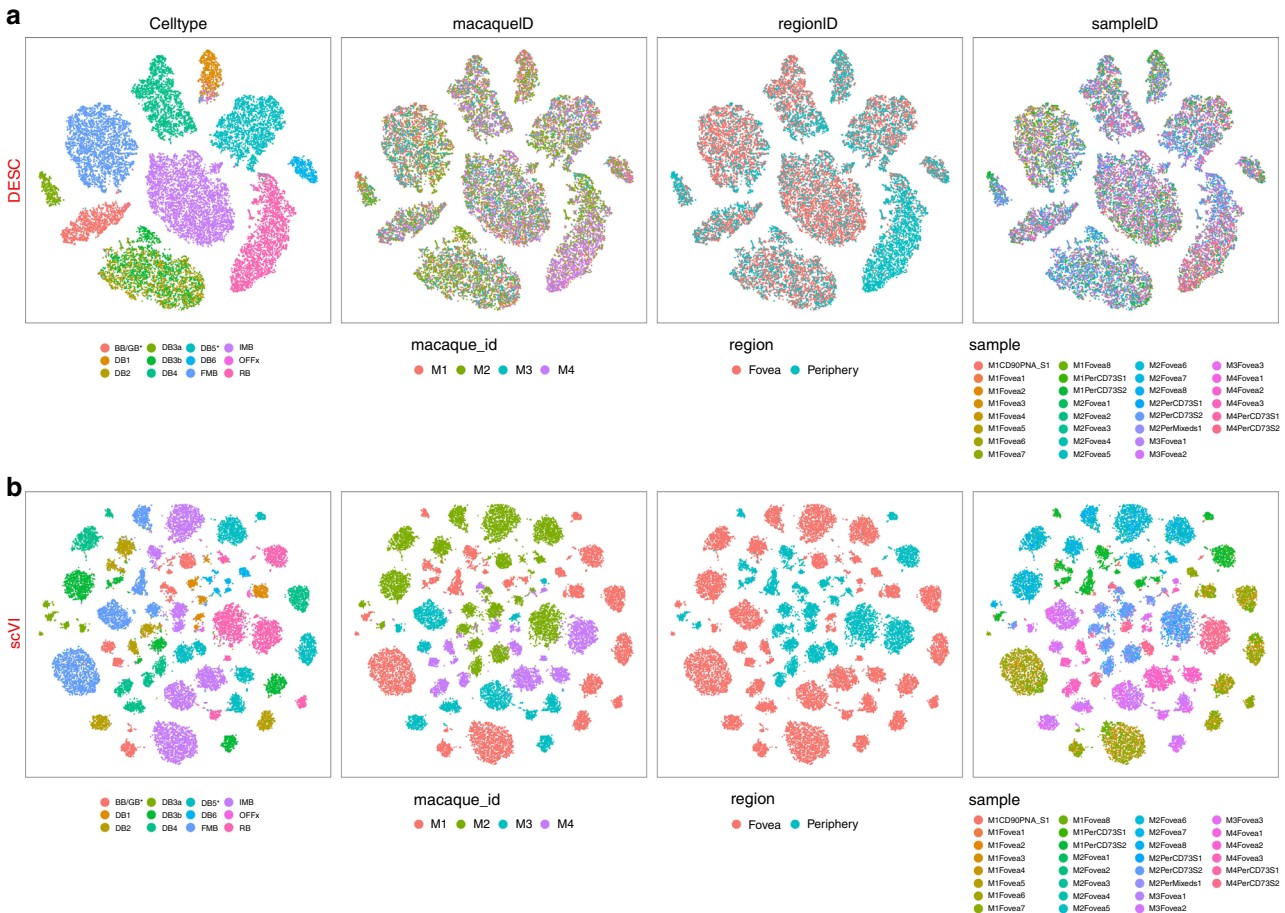

**Fig. 3 The Comparison between DESC and scVI when batch information was not provided in the analysis of macaque retina data. a** The t-SNE results of DESC when no batch information provided and the cells are colored by cell type, macaque id, region id, and sample id, respectively; **b** scVI when batch information was not provided and cells are colored by cell type, macaque id, region id, and sample id, respectively. The cells are mixed well by macaque id, region, and sample id in DESC, but are completely separated by macaque id, region, and sample id in scVI. These results indicate that DESC is able to remove complex batch effect without explicit use of batch information.

biological differences (e.g., between cell types), DESC can remove batch effect successfully. A similar assumption has been made by MNN[4].

To illustrate that DESC gradually removes batch effect over iterations, we generated t-SNE plots using the representation obtained from the bottleneck layer in DESC every two epochs for the combined pancreatic islet data. As shown in Fig. 4d, e, in the initial clustering step (before iteration starts), there is significant batch effect according to the t-SNE plot. However, during the training of the network, DESC removes batch effect gradually over iterations. Finally, after the algorithm converges, DESC returns desired result with batch effect removed in the clustering.

**Application to PBMC data with external stimulus.** To demonstrate that DESC preserves true biological variations, we considered an even more complex situation in which technical batches were completely confounded with biological conditions. This is inevitable in disease studies involving human tissues where samples need to be processed immediately to maintain cell viability, resulting in the preparation of normal and diseased samples in different batches. For data generated in such complex settings, it is desirable to remove technical batch effect while maintaining true biological variations so that disease specific subpopulations can be identified. We analyzed a dataset that includes 24,679 human PBMCs from eight patients with lupus[18] (Supplementary Note 4). The cells were split into a control group

and a matched group stimulated with interferon-beta (INF-β), which leads to a drastic but highly cell-type-specific response. This dataset is extremely challenging because removal of technical batch effect is complicated by the presence of biological differences, both between cell types under the same condition and between different conditions for the same cell type.

Since batch completely confounds with biological condition, to reduce the risk of removing true biological variations, we analyzed the data without using batch information in DESC. Figure 5a, b shows that DESC randomly mixed cells between the control and the stimulus conditions for all cell types except CD14$^+$ monocytes. Results are similar when gene expression standardization is performed within batch (Supplementary Fig. 8a). Differential expression (DE) analysis revealed a marked change in gene expression after INF-β stimulation for CD14$^+$ monocytes (Fig. 5c). The number of differentially expressed genes and the magnitude of DE, as measured by $p$ value and fold change, are several orders more pronounced than the other cell types. This is consistent with previous studies showing that CD14$^+$ monocytes have a greater change in gene expression than B cells, dendritic cells, and T cells after INF-β stimulation[19,20]. These results suggest that DESC is able to remove technical batch effect and maintain true biological variations induced by INF-β (Supplementary Figs. 9–13). Figure 5d shows the KL divergences calculated using all cells and using non-CD14$^+$ monocytes only. The KL divergence here was used to measure the degree of batch

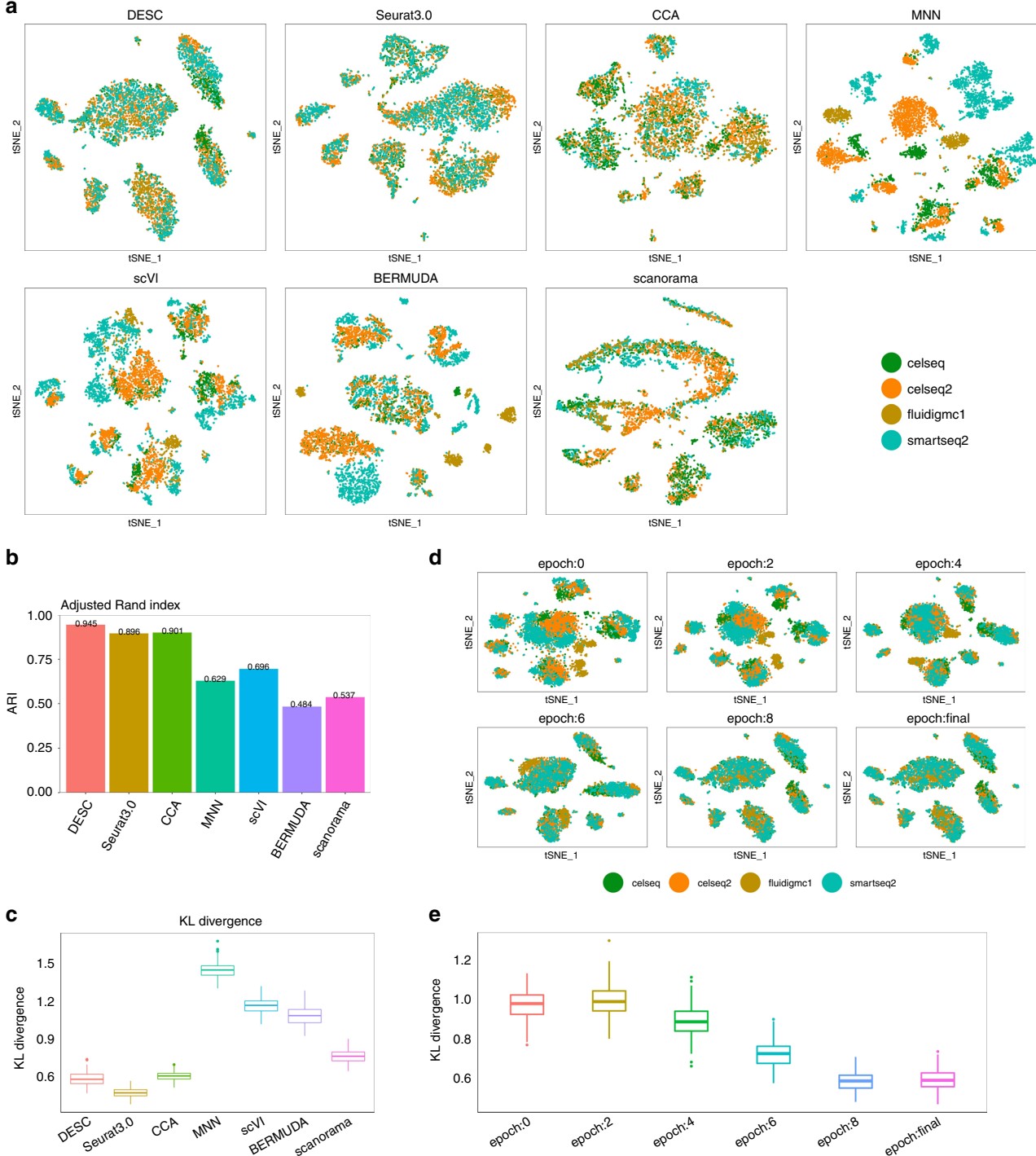

**Fig. 4 Clustering results for the pancreatic islet data generated from different scRNA-seq protocols. a** The t-SNE plots in which cells were colored by batch. **b** The ARI values of different methods. **c** The KL divergence of different methods. **d** T-SNE plots showing DESC removes batch effect gradually over iterations. **e** The KL divergence over iteration. The box in **c** and **e** represents the interquartile range, the horizontal line in the box is the median, and the whiskers represent the 1.5 times interquartile range.

effect removal (see "Methods" for evaluation metric for batch effect removal). The decreased KL divergence of DESC when $CD14^+$ monocytes were eliminated indicates that technical batch effect was effectively removed in the absence of $CD14^+$ monocytes. The KL divergences of all other methods are larger than DESC when $CD14^+$ monocytes were eliminated, indicating that they might be less effective in removing technical batch effect than DESC.

**DESC embedding preserves pseudotemporal structure**. DESC was designed to identify discrete cell clusters, but we also evaluated its performance for cells that originate from a genuinely continuous process. We analyzed a scRNA-seq dataset generated from mouse bone marrow myeloid progenitor cells[21]. Since this dataset does not have batch effect, we compared DESC with scVI only, which is also deep learning based. As shown in Fig. 6b, c, e, clustering result from DESC clearly reflects the pseudotemporal

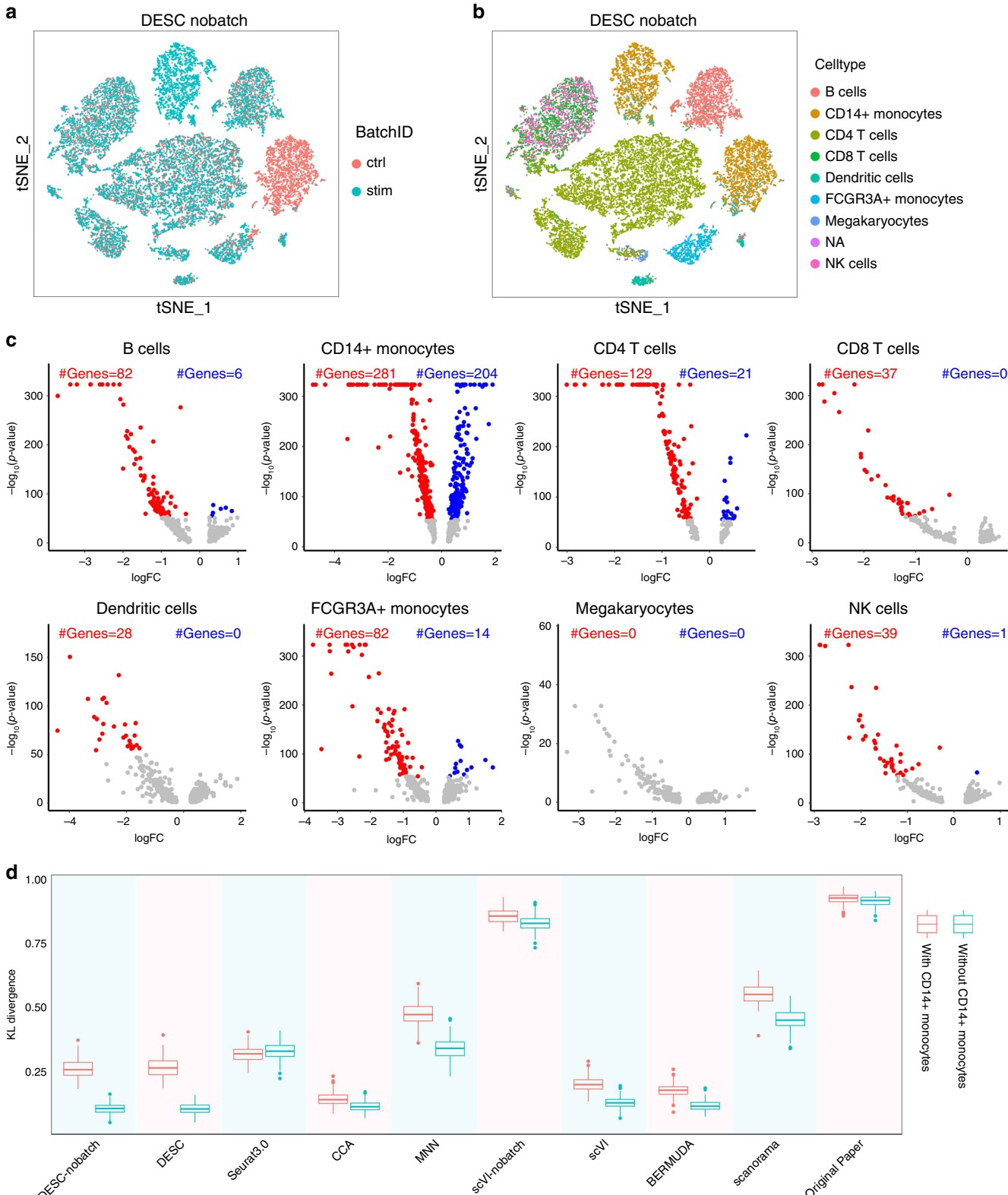

**Fig. 5 The results of PBMC data generated by Kang et al.[18]. a, b** DESC clustering without taking batch information in the analysis. **a** is colored by Batch ID and **b** is colored by cell type. **c** Volcano plots of differential expression analysis between control and stimulated conditions for each cell type. Highlighted are differential expression genes using Wilcoxon rank sum test with fold change $> e^{0.25}$ and FDR adjusted $p$ value $< 10^{-50}$. CD14$^+$ monocytes have the most number of differentially expressed genes compared with other cell types. **d** The KL divergence calculated using all cells and using non-CD14$^+$ monocytes only. The box represents the interquartile range, the horizontal line in the box is the median, and the whiskers represent the 1.5 times interquartile range.

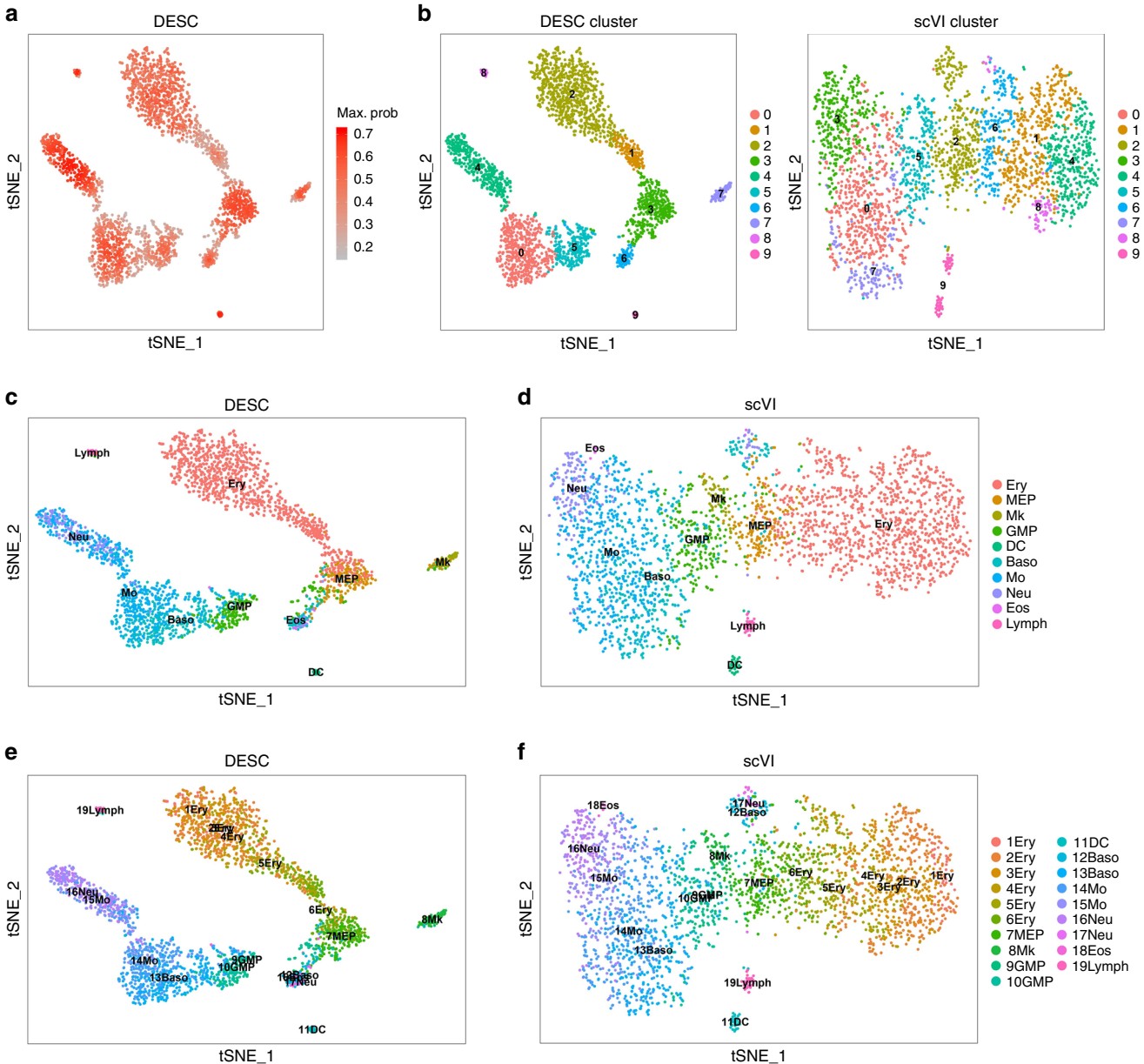

**Fig. 6 The results of mouse bone marrow data generated by Paul et al.[21]. a** The t-SNE plot showing the maximum probabilities of cluster assignments of cells. The maximum probability is the probability for the cluster that is assigned with the highest probability by DESC. **b** The t-SNE plots of clustering results by DESC and scVI. Compared with scVI, DESC yields more accurate clustering result for DC, lymph, and Mk. In scVI, the clustering result is more diffused, and Mk cells are mixed together with GMP cells. In contrast, DESC clearly separated DC, Lymph, and Mk cells from the other cell clusters. **c, d** The t-SNE plots of true cell-type labels (obtained from the original publication) for DESC and scVI. **e, f** The t-SNE plots of true cell-type labels with pseudotime ordering (obtained from the original publication) for DESC and scVI. Ery erythrocyte, MEP megakaryocyte/erythrocyte progenitors, Mk megakaryocyte, GMP granulocyte/macrophage progenitors, DC dendritic cell, Baso basophils, Mo monocyte, Neu neutrophils, Eos eosinophils; Lymph lymphocyte.

structure of the cells. DESC also identified discrete clusters in which fully differentiated cells such as dendritic cells, lymphocytes, and megakaryocytes (Mk) are well separated from cells that are still under differentiation. Interestingly, in the maximum probability plot (Fig. 6a), cells that are differentiating have relatively lower probabilities than those that are fully differentiated, suggesting that the maximum probabilities are biologically interpretable. These results indicate that DESC is able to retain both discrete and pseudotemporal structure of the cells, making it applicable to a wider range of data. In contrast, clustering result from scVI is more diffused, and Mk cells are mixed with granulocyte/macrophage progenitors (GMP), although Mk cells have distinct transcriptional profiles from GMPs[21] (Fig. 6b, d, f).

**DESC embedding removes batch effect for monocytes**. We further evaluated whether DESC is able to remove batch effect for cells that originate from a continuous process. We analyzed a scRNA-seq dataset generated from monocytes derived from human peripheral blood mononuclear cells by Ficoll separation followed by CD14- and CD16-positive cell selection. This dataset includes 10,878 monocytes collected from a single healthy subject. The cells were processed in three batches from blood drawn on three different days, sequentially 77 and 33 days apart. Although based on surface markers, monocytes can be classified as classical ($CD14^{++}/CD16^{-}$), intermediate ($CD14^{++}/CD16^{+}$), and non-classical patrolling ($CD14^{-}/CD16^{++}$) subpopulations, clustering analysis based on scRNA-seq data indicates that these cells show

continuous characteristics, making it difficult to identify discrete cell clusters.

Therefore, we used the low-dimensional representation obtained from DESC as input for Monocle3[22], to reconstruct the trajectory in this dataset. Figure 7a shows that this hybrid approach reveals a clear transitioning path across cells, and the estimated pseudotime for the three batches has similar distribution, indicating that batch effect is successfully removed (Supplementary Table 5). This is further confirmed by the small KL divergence that quantifies the degree of batch effect removal across the three batches (Fig. 7i). Compared with DESC, the other methods are less effective in removing batch effect in this setting (Fig. 7b–g, i and Supplementary Table 5), which can obscure true biological signals. For example, when using the CCA components from Seurat 3.0 as input for Monocle3, the cells were less randomly mixed, the reconstructed trajectories are more diffused, and the pseudotime distributions are noticeably different across batches (Fig. 7b). BERMUDA and scanorama produced the least similar pseudotime distributions across batches, which agrees with their relative performance for other datasets we evaluated (Fig. 7f, g).

To examine if the estimated pseudotime obtained from DESC representation is biologically interpretable, we examined expression patterns for known marker genes over the estimated pseudotime. Figure 8a shows that S100A8, a known marker gene for classical monocytes, has increased expression for cells as the estimated pseudotime from DESC increases, indicating that the cells are gradually changing from nonclassical to intermediate, and then to classical monocytes. Moreover, the estimated gene expression curves are similar across the three batches. In contrast, when using uncorrected raw gene expression data as input for Monocle3, the three batches show drastically different expression patterns over the estimated pseudotime, suggesting that failure to remove batch effect can lead to misinterpretation of the data (Fig. 8h). The other methods we evaluated also showed consistent gene expression patterns over pseudotime across the three batches, but with unexpected shapes (Fig. 8b–g). For example, for MNN, S100A8 has high expression when pseudotime is near 0.75 (Fig. 8d), however, as a marker gene for classical monocytes, its expression is expected to gradually increase when the cells are moving away from nonclassical to intermediate, and then to classical monocytes.

These analyses have important implications for studies that involve time-course data for samples with continuous processes in which samples at different time points are generated in separate batches. Removal of batch effect is critical for distinguishing biological and technical batch difference because failure to remove batch effect can lead to complete separation of pseudotime distributions across time points.

**Comparison of running time and memory usage**. To verify DESC is scalable for large dataset, we analyzed a 1.3 million mouse brain cell dataset generated by 10X Genomics. With the aid of NVIDIA TITAN Xp GPU, DESC finished the analysis in about 3.5 h using less than 10 GB memory (Supplementary Figs. 16 and 17 and Supplementary Note 7).

To evaluate how running time and memory usage change as the number of cells and number of batches vary, we randomly selected different number of cells from datasets with two batches, four batches, and 30 batches, respectively. For the two batch scenario, we used the data from Kang et al.[18] with CTRL and STIM defined as two different batches, and we randomly selected 1000, 2000, 5000, 8000, 10,000, and 20,000 cells from the original 24,679 cells, to evaluate the impact of the number of cells. For the four-batch and 30-batch scenarios, we used the data from Peng et al.[10] in which there are four batches if macaque id was taken as

batch definition, and 30 batches if sample id was taken as batch definition. For both scenarios, we randomly selected 1000, 2000, 5000, 8000, 10,000, 20,000, and 30,000 cells from the original 30,302 cells, to evaluate the impact of the number of cells.

Our results indicate that the running times of CCA, Seurat 3.0, and MNN increase exponentially as the number of cells and the number of batches increase (Fig. 9). Moreover, Seurat 3.0 and BERMUDA threw out an error when the number of cells within a batch is small and stopped to run (Fig. 9g, h, i). In contrast, the memory usage and running time of DESC are not affected by the number of batches, and increase linearly as the number of cells increases. It is also worth noting that scVI and BERMUDA's computing time increases substantially when the number of batches increases from two to 30 even when the number of cells remains the same. More specifically, when 30 samples were considered as batch definition in the macaque retina dataset (i.e., 30 batches), the computing time for analysis of 30,000 cells is about 0.3 h for DESC, 9.5 h for MNN, 10.4 h for scVI, 11.2 h for BERMUDA, and 0.4 h for scanorama, respectively. Furthermore, BERMUDA requires more than 32 GB memory even for a dataset that only has 30,000 cells when the number of batches is 30. Methods such as CCA and Seurat 3.0 broke due to memory issue and failed to produce results when the number of batches is 30. These computing constraints will limit the practical utility of these methods in large-scale single-cell studies, especially for studies with many batches.

All data analyses reported in this section were conducted on Ubuntu 18.04.1 LTS with Intel® Core (TM) i7-8700K CPU @ 3.70 GHz and 64 GB memory, except for the 1.3 million cells mouse brain data. For the 1.3 million cells mouse brain dataset, we analyzed it on Ubuntu 16.04.4 LTS with Intel(R) Xeon(R) CPU E5-2620 v4 @ 2.10 GHz and total 128 GB memory.

## Discussion

In summary, we have presented a deep learning based algorithm that clusters scRNA-seq data by iteratively optimizing a clustering objective function with a self-training target distribution. Through iterative learning, DESC removes complex batch effect and maintains true biological variations. As a soft clustering based algorithm, the cluster assignment probabilities from DESC are biologically interpretable and can reveal both discrete and pseudotemporal structure of cells. We have performed comprehensive evaluations and compared with many state-of-the-art methods for scRNA-seq analysis. Figure 10 shows a summary of the performance of each evaluated method. Overall, DESC achieved high clustering accuracy across all datasets and showed stable results under a wide range of scenarios.

ScRNA-seq has emerged as a revolutionary tool in biomedical research. In the last few years, studies employing scRNA-seq technique have grown exponentially. Despite its promises, many computational challenges must be overcome before we can fully reap the benefit of scRNA-seq. A critical first step in many scRNA-seq studies is to cluster cells into biologically meaningful entities. Effective removal of technical batch effect is important to ensure the validity and interpretability of clustering results. Through comprehensive evaluations with many datasets, we have shown that DESC offers a proper balance of clustering stability and accuracy, small footprint on memory usage, does not explicitly require batch definition for batch effect removal, and can speed up computation by utilizing GPU when available. As the scale of single-cell studies continues to grow, we believe that DESC will be a valuable tool for biomedical researchers to better disentangle complex cellular heterogeneity, and will facilitate the translation of basic research findings into clinical studies of human disease.

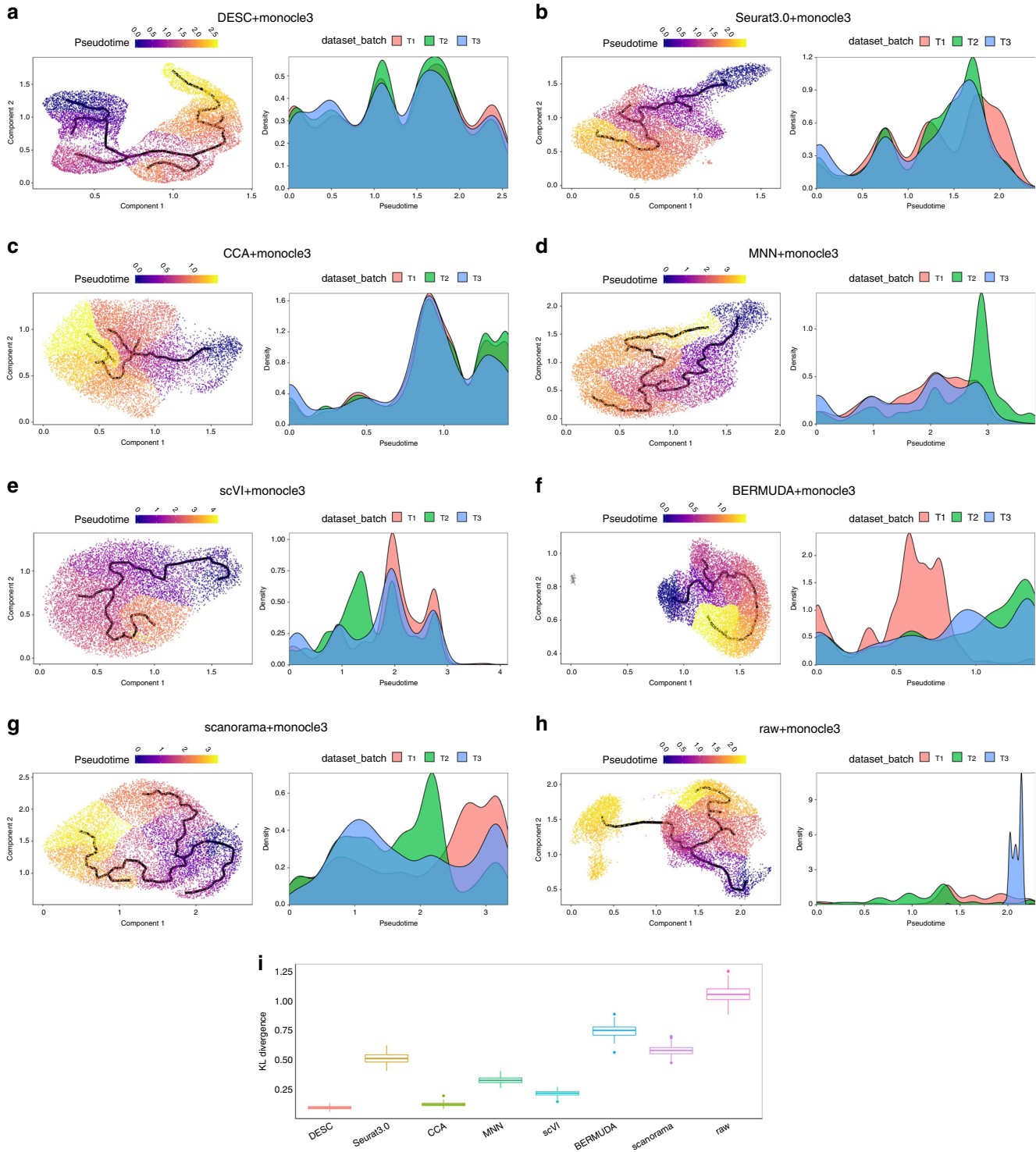

**Fig. 7 The estimated pseudotime plots for the human monocyte data.** Shown are the results of Monocle3 estimated pseudotime using **a** low-dimensional representation from DESC as input; **b** CCA components from Seurat 3.0 as input; **c** CCA components from method CCA as input; **d** PCA components of corrected gene expression values from MNN as input; **e** low-dimensional representation from scVI as input; **f** low-dimensional representation from BERMUDA as input; **g** low-dimensional representation from scanorama as input; **h** raw gene count matrix as input. Default parameters in Monocle3 were used to conduct dimension reduction and pseudotime estimation. **i** KL divergences that measure of the degree of batch effect removal for different methods. "Raw" represents the output of Monocle3 using the raw gene count matrix as the input. The box represents the interquartile range, the horizontal line in the box is the median, and the whiskers represent the 1.5 times interquartile range.

## Methods

**The DESC algorithm**. Analysis of scRNA-seq data often involves clustering of cells into different clusters and selection of highly variable genes for cell clustering. As these are closely related, it is desirable to use a data driven approach to cluster cells and select genes simultaneously. This problem shares similarity with pattern

recognition, in which clear gains have resulted from joint consideration of the classification and feature selection problems by deep learning. However, for scRNA-seq data, a challenge is that we cannot train deep neural network with labeled data as cell-type labels are typically unknown. To solve this problem, we take inspiration from recent work on unsupervised deep embedding for clustering

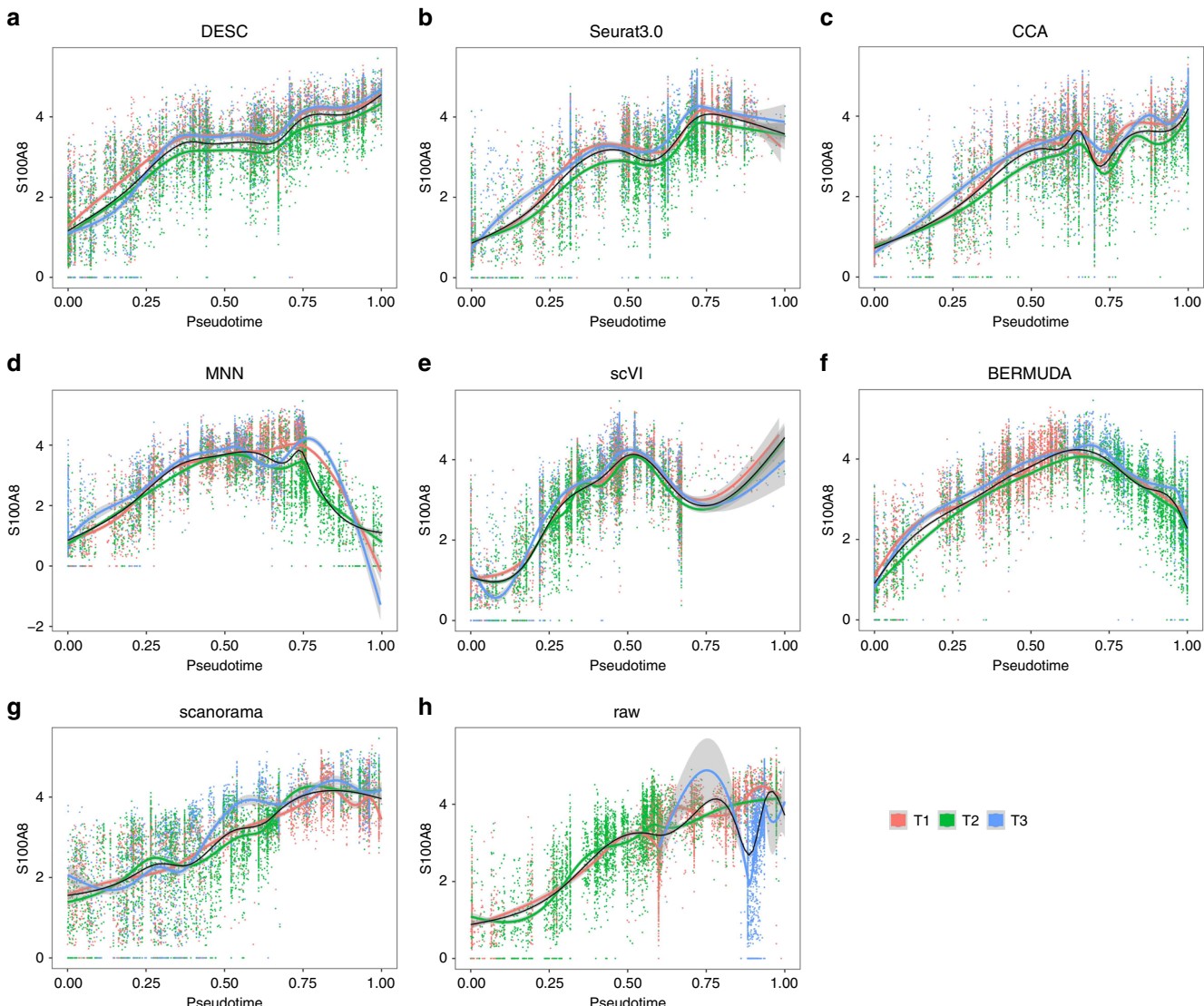

**Fig. 8 The expression of marker gene S100A8 (classical monocytes) over pseudotime for different methods for cells across batches.** The black line is the smoothed expression curve when cells from all batches are included. The red, green, and blue lines are the smoothed expression curves for cells from T1, T2, and T3, respectively. Pseudotime from all methods was scaled to [0, 1] for comparison. **a** Low-dimensional representation from DESC as input; **b** CCA components from Seurat 3.0 as input; **c** CCA components from method CCA as input; **d** PCA components of corrected gene expression values from MNN as input; **e** low-dimensional representation from scVI as input; **f** low-dimensional representation from BERMUDA as input; **g** low-dimensional representation from scanorama as input; **h** raw gene count matrix as input.

analysis[23], in which we iteratively refine clusters with an auxiliary target distribution derived from the current soft cluster assignment. This self-learning process gradually improves clustering as well as feature representation. As shown by our comprehensive evaluations across a wide range of datasets, this procedure also gradually removes batch effect over iterations.

*Overview of DESC*: The DESC procedure starts with parameter initialization, in which a stacked autoencoder is used for pretraining and learning low-dimensional representation of the input gene expression matrix. The corresponding encoder is then added to the iterative clustering neural network. The cluster centers are initialized by the Louvain's clustering algorithm[6], which aims to optimize modularity for community detection. This clustering returns data in a feature space that allows us to obtain centroids in the initial stage of the iterative clustering. Below, we describe each component of the DESC procedure in detail.

*Parameter initialization by stacked autoencoder*: Let $\mathbf{X} \in R^{n \times p}$ be the gene expression matrix obtained from a scRNA-seq experiment, in which rows correspond to cells and columns correspond to genes. Due to sparsity and high dimensionality of scRNA-seq data, to perform clustering, it is necessary to transform the data from high dimensional space $R^p$ to a lower dimensional space $R^d$ in which $d \ll p$. Traditional dimension-reduction techniques, such as principal component analysis, operate on a shallow linear embedded space, and thus have limited ability to represent the data. To better represent the data, we perform

feature transformation by a stacked autoencoder, which have been shown to produce well-separated representations on real datasets.

The stacked autoencoder network is initialized layer by layer with each layer being an autoencoder trained to reconstruct the previous layer's output. After layer-wise training, all encoder layers are concatenated, followed by all decoder layers, in reverse layer-wise training order. The resulting autoencoder is then fine-tuned to minimize reconstruction loss. The final result is a multilayer autoencoder with a bottleneck layer in the middle. After fine tuning, the decoder layers are discarded, and the encoder layers are used as the initial mapping between the original data space and the dimension-reduced feature space, as shown in Fig. 1a.

Since the number of true clusters for a scRNA-seq dataset is typically unknown, we apply the Louvain's method, a graph-based method that has been shown to excel over other clustering methods, on the feature space $\mathbf{Z}$ obtained from the bottleneck layer. This analysis returns the number of clusters, denoted by $K$, and the corresponding cluster centroids $\{\mu_j : j = 1, \ldots, K\}$, which will be used as the initial clustering for DESC.

*Iterative clustering*: After cluster initialization, we improve the clustering using an unsupervised algorithm that alternates between two steps until convergence. In the first step, we compute a soft assignment of each cell between the embedded points and the cluster centroids. Following van der Maaten and Hinton[24], we use the Student's $t$ distribution as a kernel to measure the similarity between embedded

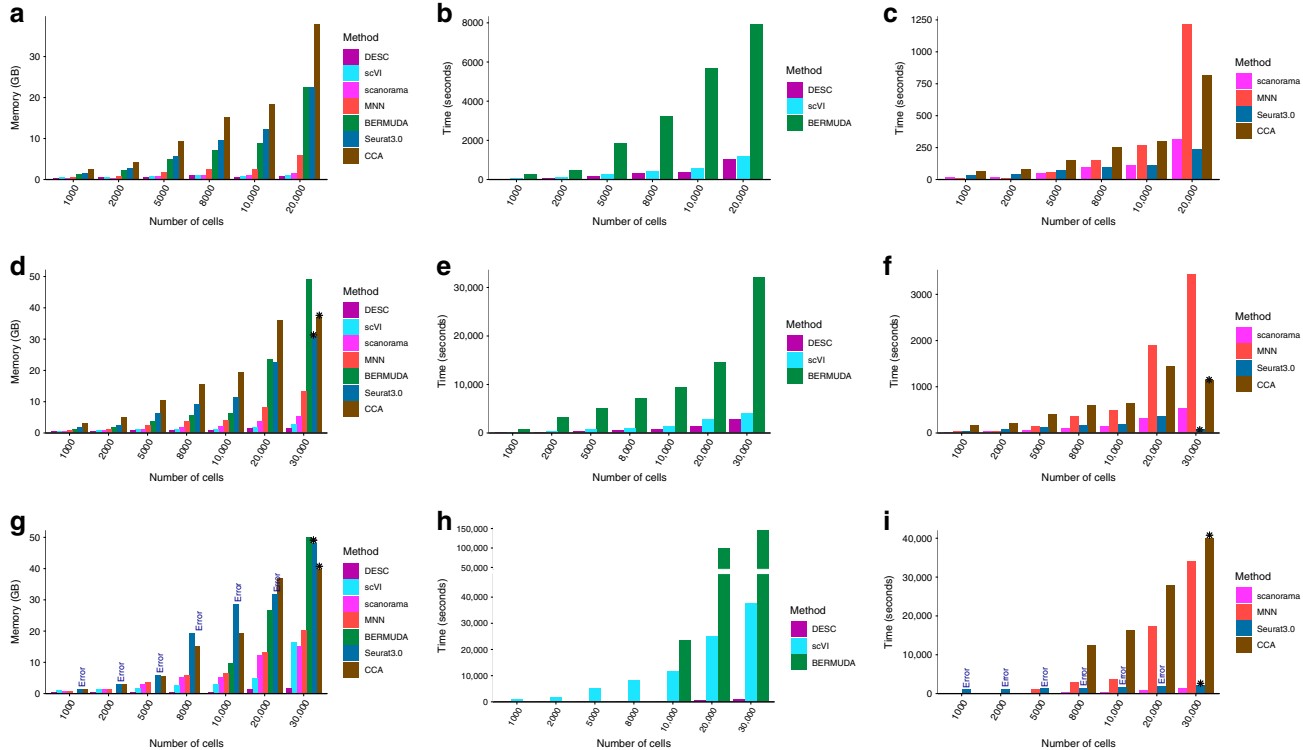

**Fig. 9 Comparison of memory usage (first column) and running time (second and third columns). a–c** The number of batches for analyzed samples is 2. Analyzed data were from Kang et al.[18]. **d–f** The number of batches for analyzed samples is 4. **g–i** The number of batches for analyzed samples is 30. For **d–i** the analyzed data were form Peng et al.[10] in which there are four batches when taking macaque id as batch definition and 30 batches when taking sample as batch definition. Because DESC, scVI, and BERMUDA are deep learning based methods, we put them together for ease of comparison. Remark: the reason that the running time of batch = 30 is smaller than that of batch = 4 for DESC is because when the data were standardized by sample id (i.e., when batch = 30), the algorithm converged quickly before reaching to the maximum number of epochs (300). The "Error" in the bar plot in **g** and **i** indicates that there was an error when using Seurat 3.0. This is because the numbers of cells in some batches are very small. The "asterisk" above the bar plot in **d**, **f**, **g**, and **i** indicates that the corresponding method broke due to memory issue (i.e., cannot allocate memory). Therefore, the recorded time is the computing time until the method broke. When the number of batches is 30, BERMUDA always throws out an error when the number of cells is less than 8000, so we only report BERMUDA when the number of cells ≥ 10,000. In addition, the reported running time and memory usage only include clustering procedure and not include the procedure of computing t-SNE or UAMP. All reported time and memory usage related to this figure were analyzed on our workstation Ubuntu 18.04.1 LTS with Intel® Core(TM) i7-8700K CPU @ 3.70 GHz and 64 GB memory.

| Method | Clustering stability and accuracy | Pairwise analysis | Require batch information | Memory usage with increasing number of cells | Whether utilize GPU | Programming language |
|---|---|---|---|---|---|---|
| DESC | | No | No | | Yes | python |
| scVI | | No | Yes | | Yes | python |
| scanorama | | Yes | Yes | | No | python |
| MNN | | Yes | Yes | | No | python |
| BERMUDA | | No | Yes | | Yes | R+python |
| Seurat3.0 | | Yes | Yes | | No | R |
| CCA | | Yes | Yes | | No | R |

**Fig. 10 Comparison with different methods for batch effect removal.** The second column was computed based on results shown in Fig. 1e, and the error bar is the standard error of ARI when using different information as batch in analysis. The fifth column was computed based on results shown in Fig. 9a. For each method, memory usage was shown for 1000, 2000, 5000, 8000, 10,000, and 20,000 cells, respectively.

point $z_i$ for cell $i$ and centroid $\mu_j$ for cluster $j$

$$q_{ij} = \frac{\left(1 + \|z_i - \mu_j\|^2 / \alpha\right)^{-1}}{\sum_{j'} \left(1 + \|z_i - \mu_{j'}\|^2 / \alpha\right)^{-1}}, \quad (1)$$

where $z_i = f_W(x_i) \in \mathbf{Z}$ corresponds to $x_i \in \mathbf{X}$ after embedding, and $\alpha$ is the degree of freedom of the Student's $t$ distribution. We also evaluated the performance of Gaussian distribution as a kernel, but found it to be less robust than the $t$ distribution (Supplementary Note 3).

In the second step, we refine the clusters by learning from cells with high confidence cluster assignments with the help of an auxiliary target distribution.

Specifically, we define the objective function as a KL divergence loss between the soft cell assignments $q_i$ and the auxiliary distribution $p_i$ for cell $i$ as

$$L = KL(\mathbf{P} \parallel \mathbf{Q}) = \sum_{i=1}^{n} \sum_{j=1}^{K} p_{ij} \log \frac{p_{ij}}{q_{ij}}, \qquad (2)$$

where the auxiliary distribution $\mathbf{P}$ is defined as

$$p_{ij} = \frac{q_{ij}^2 / \sum_{i=1}^{n} q_{ij}}{\sum_{j=1}^{K} \left( q_{ij}^2 / \sum_{i=1}^{n} q_{ij} \right)}. \qquad (3)$$

The encoder is fine-tuned by minimizing $L$ in Eq. (2) iteratively. The above definition of the auxiliary distribution $\mathbf{P}$ can improve cluster purity by putting more emphasis on cells assigned with high confidence. Given that the target distribution $\mathbf{P}$ is defined by $\mathbf{Q}$, minimizing $L$ implies a form of self-training. Also, $p_{ij}$ gives the probability of cell $i$ that belongs to cluster $j$, and this probability can be used to measure the confidence of cluster assignment for each cell. Because $\alpha$ is insensitive to the clustering result, we let $\alpha = 1$ for all datasets by default.

*Optimization of the KL divergence loss*: We jointly optimize the cluster centers $\{\mu_j : j = 1, \ldots, K\}$ and the deep neural network parameters using stochastic gradient descent. The gradients of $L$ with respect to feature space embedding of each data point $z_i$ and each cluster center $\mu_j$ are

$$\frac{\partial L}{\partial z_i} = \frac{\alpha + 1}{\alpha} \sum_{j=1}^{K} \left( 1 + \frac{z_i - \mu_j^2}{\alpha} \right)^{-1} \times (p_{ij} - q_{ij})(z_i - \mu_j), \qquad (4)$$

$$\frac{\partial L}{\partial \mu_j} = \frac{-(\alpha + 1)}{\alpha} \sum_{i=1}^{n} \left( 1 + \frac{z_i - \mu_j^2}{\alpha} \right)^{-1} \times (p_{ij} - q_{ij})(z_i - \mu_j). \qquad (5)$$

These gradients are then passed down to the deep neural network and used in standard backpropagation to compute the deep neural network's parameter gradient. We use Keras to train our model. During each iteration, i.e., when loss is not decreasing or the epoch number threshold is reached, we update the auxiliary distribution $\mathbf{P}$, and optimize cluster centers and encoder parameters with the new $\mathbf{P}$. This iterative procedure stops when the proportion of cells that changes cluster assignment between two consecutive steps is less than tol. Specifically, tol is calculated as tol $= \frac{\# \left| Y_{\text{curr}} \neq Y_{\text{prev}} \right|}{n}$, where $Y_{\text{curr}}$ is the cluster id obtained by the maximum cluster assignment probability in the current step, $Y_{\text{prev}}$ is the corresponding cluster id in the previous step, $n$ is the total number of cells, and $\# \left| Y_{\text{curr}} \neq Y_{\text{prev}} \right|$ is the number of cells in which $Y_{\text{curr}}$ does not agree with $Y_{\text{prev}}$. We let tol $= 0.005$ by default.

*Architecture of the deep neural network in DESC*: Depending on the number of cells in the dataset, we suggest different numbers of hidden layers and different numbers of nodes in the encoder. Supplementary Table 2 gives the default numbers of hidden layers and nodes in DESC.

DESC allows users to specify their own numbers of hidden layers and nodes. We recommend using more hidden layers and more nodes per layer for datasets with more cells so that the complexity of the data can be captured by the deep neural network. We use ReLU as the activation function except for the bottleneck layer and last decoder layer, in which tanh is used as the activation function. The reason why we use tanh is that we must guarantee the values in feature representation and output of decoder range from negative to positive. The default hyperparameters for the autoencoder are listed in Supplementary Table 3.

**Data normalization and gene selection**. The normalization involves two steps. In the first step, cell level normalization is performed, in which the UMI count for each gene in each cell is divided by the total number of UMIs in the cell, multiplied by 10,000, and then transformed to a natural log scale. In the second step, gene level normalization is performed in which the cell level normalized values for each gene are standardized by subtracting the mean across all cells and divided by the standard deviation across all cells for the given gene. When batch information is provided, gene expression standardization is performed across cells in each batch separately. Highly variable genes are selected using the filter_genes_dispersion function from the Scanpy package[25] (https://github.com/theislab/scanpy).

**Evaluation metric for clustering**. For published datasets in which the reference cell-type labels are known, we use ARI to compare the performance of different clustering algorithms. Larger values of ARI indicate higher accuracy in clustering. The ARI can be used to calculate similarity between the clustering labels obtained from a clustering algorithm and the reference cluster labels. Given a set of $n$ cells and two sets of clustering labels of these cells, the overlap between the two sets of clustering labels can be summarized in a contingency table, in which each entry denotes the number of cells in common between the two sets of clustering labels.

Specifically, the ARI is calculated as

$$ARI = \frac{\sum_{jj'} \binom{n_{jj'}}{2} - \left[ \sum_j \binom{a_j}{2} \sum_{j'} \binom{b_{j'}}{2} \right] / \binom{n_{jj'}}{2}}{\frac{1}{2} \left[ \sum_j \binom{a_j}{2} + \sum_{j'} \binom{b_{j'}}{2} \right] - \left[ \sum_j \binom{a_j}{2} \sum_{j'} \binom{b_{j'}}{2} \right] / \binom{n_{jj'}}{2}}, \qquad (6)$$

where $n_{jj'}$ is the number of cells assigned to cluster $j$ based on the reference cluster labels, and cluster $j'$ based on clustering labels obtained from a clustering algorithm, $a_j$ is the number of cells assigned to cluster $j$ in the reference set, and $b_{j'}$ is the number of cells assigned to cluster $j'$ by the clustering algorithm.

**Evaluation metric for batch effect removal**. We use KL divergence to evaluate the performance of various single-cell clustering algorithms for batch effect removal, i.e., how randomly are cells from different batches mixed together within each cluster. The KL divergence of batch mixing for $B$ different batches is calculated as

$$KL = \sum_{b=1}^{B} p_b \log \frac{p_b}{q_b}, \qquad (7)$$

where $q_b$ is the proportion of cells from batch $b$ among all cells, and $p_b$ is the proportion of cells from batch $b$ in a given region based on results from a clustering algorithm, with $\sum_{b=1}^{B} q_b = 1$ and $\sum_{b=1}^{B} p_b = 1$. We calculate the KL divergence of batch mixing by using regional mixing KL divergence defined above using 100 randomly chosen cells from all batches. The regional proportion of cells from each batch is calculated based on the set of $K$ nearest neighboring cells from each randomly chosen cell ($K$ can be set differently according the number of batches, we suggest $K = 5 \times$ the number of batches). The final KL divergence is then calculated as the average of the regional KL divergence. We repeated this procedure for 200 iterations with different randomly chosen cells to generate box plots of the final KL divergence. Smaller final KL divergence indicates better batch mixing, i.e., more effective in batch effect removal.

**Reporting summary**. Further information on research design is available in the Nature Research Reporting Summary linked to this article.

## Data availability

We analyzed multiple scRNA-seq datasets. The published datasets used in this manuscript are available through the following websites or accession numbers: (1) bipolar cells from macaque retina (GSE118480); (2) human pancreatic islet data: CelSeq (GSE81076), CelSeq2 (GSE85241), Fluidigm C1 (GSE86469), and SMART-Seq2 (E-MTAB-5061); (3) mouse hematopoietic stem cells with bone marrow (GSE727857), or can be downloaded using command scanpy.datasets.paul15() in python module scanpy; (4) human PBMC data (GSE96583); and (5) mouse brain data by 10X Genomics can be downloaded from https://support.10xgenomics.com/single-cell-gene-expression/datasets/1.3.0/1M_neurons. The human monocyte data generated by us can be downloaded from GEO (GSE146974).

Macaque retina dataset. The data were generated by Peng et al.[10] in which 165,679 cells were generated using Drop-seq, including 42,020 retinal ganglion cells, 36,268 nonneuronal cells, 30,302 bipolar cells, 30,236 amacrine cells, 24,707 photoreceptors, and 2146 horizontal cells, but here we only focus on the 30,302 bipolar cells. This dataset allows us to examine batch effect at the different level (sample, animal, and region).

Human pancreatic islet datasets. We chose human pancreatic islet scRNA-seq datasets generated using different scRNA-seq protocols, including CelSeq (GSE81076, 1004 cells)[16], CelSeq2 (GSE85241, 2285 cells)[17], Fluidigm C1 (GSE86469, 638 cells)[14], and SMART-Seq2 (E-MTAB-5061, 2394 cells)[15] and the total number of cells in the combined dataset is 6321.

Human PBMC dataset. The data were generated by Kang et al.[18] in which 24,679 PBMC cells were obtained and processed from eight patients with lupus using 10X. These cells were split into two groups: one stimulated with INF-β and a culture-matched control. This dataset allows us to examine whether technical batch effect can be removed in the presence of true biological variations.

Mouse bone marrow myeloid progenitor cell dataset. This dataset was generated by Paul et al.[21], which includes 2730 cells from multiple progenitor subgroups showing unexpected transcriptional priming towards seven differentiation fates. This dataset allows us to examine whether DESC can reveal pseudotemporal structure of the cells.

Human monocyte dataset. The data were generated by our group in which 10,878 monocytes derived from blood were obtained from one healthy human subject. The cells were processed in three batches from blood drawn on three different days, sequentially 77 and 33 days apart. Briefly, monocytes were isolated from freshly collected human peripheral blood mononuclear cells by Ficoll separation followed by CD14- and CD16-positive cell selection. This dataset allows us to examine whether DESC is able to remove batch effect while retaining pseudotemporal structure of the cells.

1.3 million brain cells from E18 mice. This dataset was downloaded from the 10X Genomics website. It includes 1,306,127 cells from cortex, hippocampus, and subventricular zone of two E18 C57BL/6 mice.

A complete list of the datasets analyzed in this paper is provided in Supplementary Table 1.

## Code availability

An open-source implementation of the DESC algorithm and code to reproduce the results can be downloaded from https://eleozzr.github.io/desc/.

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

## Acknowledgements

This work was supported by the following grants: NIH R01GM108600 (to M.L.), R01GM125301 (to M.L.), R01EY030192 (to M.L.), R01HL113147 (to M.L. and M.P.R.), and R01HL150359 (to M.L. and M.P.R.).

## Author contributions

This study was conceived of and led by M.L. and G.H. Authors X.L., G.H., and M.L. designed the model and algorithm. X.L. implemented the DESC software and led the data analysis with input from M.L., G.H., K.W., Y.L., J.Z., D.S., K.S., and M.P.R. Authors W.K. and Y.L. helped with software development and testing. M.P.R. and H.P. generated the human monocyte scRNA-seq data, and provided input on the monocyte data analysis. D.S. provided input on the macaque retina scRNA-seq data analysis. M.L., X.L., and G.H. wrote the paper with feedback from K.W., Y.L., H.P., J.Z., D.S., K.S., and M.P.R.

## Competing Interests

The authors declare no competing interests.
