## [Peer Review File · Nature Communications]

REVIEWERS' COMMENTS:

Reviewer #2 (Remarks to the Author):

The authors addressed the issues I raised in the previous review. Here are a few minor comments.

1. The authors used ARI to measure clustering accuracy which is an agreement between clustering result and some reference class label. It would be easier for the reader if the authors add, in the text, for each data set evaluated, what reference labels are and how are the labels obtained. Or, maybe expand Supplementary Table 1 to include the cell types and composition so that the readers can have an easy appreciation of the complexity of the data. For example, in the macaque retina data, this information is not provided in the text. I found it in Fig3a. The cell types in the pancreatic data seem to be only available in Supp Fig 6. In many cases the reference labels were given by the authors of the original data. It would be helpful if the authors could simply add a sentence about how these were obtained – using known cell type markers to cluster, sorted by cell surface markers, or ?

2. For the classification accuracy mentioned on Page 3,
“47 Although Seurat 3.0 also has high ARI (0.896), its classification
48 accuracies for α and β cells are only 92.1% and 88.1%, respectively, whereas the accuracies are
49 96.5% for α cells and 98.3% for β cells in DESC, respectively (Supplementary Note 3).” Can you clarify
what is classification accuracy? Is it the proportion of true alpha cells identified as alpha cells ($P(\text{true } \alpha$
|classified as α) similar to sensitivity) or the proportion of cells classified as alpha cells that are indeed alpha
cells (similar to positive predictive value)?

Figure legends

Please add information about Figure 3: Are these tSNE plots too? These do not resemble any of the tSNE plots in Fig2b, though both are on macaque data. I suppose Fig 3a are still tSNE plots except that batch information is blinded?

Reviewer #4 (Remarks to the Author):

In this manuscript, the authors have presented a novel method based on unsupervised deep embedding algorithm to perform clustering in single-cell RNAseq data while simultaneously removing batch effects. The manuscript is well organized and well written. The method is novel and well justified in the context applied. Extensive evaluations using real datasets demonstrate not only the overall superior performance in terms of clustering accuracy, but also the stability and robustness to various realistic complications to complex and a continuum mixture of cell types, cell states, systematic differences in experimental protocols and additional batch effects. The authors have carefully dealt with all the previous comments.

I would specifically want to comment on the first and presumably the most important comment from reviewer 1 regarding little improvement over existing methods such as MNN or CCA. While I agree that in some settings, DESC is comparable but the fact that DESC is always the best or close to the best, robust under various realistic scenarios, and computationally efficient, justifying adding DESC to the standard toolbox for dealing with single cell RNAseq data.

Finally and importantly, the capability to analyze all data simultaneously rather than pairwise analysis is another commendable advantage over MNN and Seurat.

We thank the two reviewers for their constructive feedback. Below are our point-by-point responses to the reviewers' comments. The original reviewers' comments are in **bold italics** and our responses are in normal font colored in blue.

Reviewer #2 (Remarks to the Author):

The authors addressed the issues I raised in the previous review. Here are a few minor comments.

1. The authors used ARI to measure clustering accuracy which is an agreement between clustering result and some reference class label. It would be easier for the reader if the authors add, in the text, for each data set evaluated, what reference labels are and how are the labels obtained. Or, maybe expand Supplementary Table 1 to include the cell types and composition so that the readers can have an easy appreciation of the complexity of the data. For example, in the macaque retina data, this information is not provided in the text. I found it in Fig3a. The cell types in the pancreatic data seem to be only available in Supp Fig 6. In many cases the reference labels were given by the authors of the original data. It would be helpful if the authors could simply add a sentence about how these were obtained – using known cell type markers to cluster, sorted by cell surface markers, or ?

Thanks for your suggestions. We have expanded Supplementary Table 1 to include cell type names and the corresponding number of cells in each cell type so that the readers can have an easy understanding of the data.

2. For the classification accuracy mentioned on Page 3,

“Although Seurat 3.0 also has high ARI (0.896), its classification accuracies for α and β cells are only 92.1% and 88.1%, respectively, whereas the accuracies are 96.5% for α cells and 98.3% for β cells in DESC, respectively (Supplementary Note 3).” Can you clarify what is classification accuracy? Is it the proportion of true alpha cells identified as alpha cells ($P(\text{true } \alpha | \text{classified as } \alpha)$ similar to sensitivity) or the proportion of cells classified as alpha cells that are indeed alpha cells (similar to positive predictive value)?

The classification accuracy here refers to the proportion of true alpha cells that were identified as alpha cells. The true cell type labels were obtained from the original papers.

Figure legends

Please add information about Figure 3: Are these tSNE plots too? These do not resemble any of the tSNE plots in Fig2b, though both are on macaque data. I suppose Fig 3a are still tSNE plots except that batch information is blinded?

Yes, plots in Figure 3 are tSNE plots in which the coordinates are based on DESC (Fig 3a) and scVI (Fig 3b), respectively, when batch information was not provided in the analysis of the macaque retina data. We have clarified this point in the legend of Figure 3.

Reviewer #4 (Remarks to the Author):

In this manuscript, the authors have presented a novel method based on unsupervised deep embedding algorithm to perform clustering in single-cell RNAseq data while simultaneously removing batch effects. The manuscript is well organized and well written. The method is novel and well justified in the context applied. Extensive evaluations using real datasets demonstrate not only the overall superior performance in terms of clustering accuracy, but also the stability and robustness to various realistic complications to complex and a continuum mixture of cell types, cell states, systematic differences in experimental protocols and additional batch effects. The authors have carefully dealt with all

the previous comments.

I would specifically want to comment on the first and presumably the most important comment from reviewer 1 regarding little improvement over existing methods such as MNN or CCA. While I agree that in some settings, DESC is comparable but the fact that DESC is always the best or close to the best, robust under various realistic scenarios, and computationally efficient, justifying adding DESC to the standard toolbox for dealing with single cell RNAseq data.

Finally and importantly, the capability to analyze all data simultaneously rather than pairwise analysis is another commendable advantage over MNN and Seurat.

Thanks for your positive comments about this paper.